# Assessing the Pedological Impact of Local Anesthesia Dental Simulator as Serious Game

Sobia Zafar [1,†], Kristina Mladenovic [2], Sakher AlQahtani [3], Chaitanya Puranik [4] and Rasa Mladenovic [5,*,†]

1   School of Dentistry, The University of Queensland, Brisbane 4001, Australia; s.zafar@uq.edu.au
2   Center for Rehabilitation Medicine, University Clinical Center Kragujevac, Faculty of Medical Sciences, University of Kragujevac, 34000 Kragujevac, Serbia; kristinamladenovic1990@gmail.com
3   College of Dentistry, King Saud University, Riyadh 11564, Saudi Arabia; s.alqahtani@uq.edu.au
4   Department of Pediatric Dentistry, Children's Hospital Colorado and School of Dental Medicine, University of Colorado Anschutz Medical Campus, Aurora, CO 80010, USA; chaitanya.puranik@childrenscolorado.org
5   Department of Dentistry, Faculty of Medical Sciences, University of Kragujevac, 34000 Kragujevac, Serbia
*   Correspondence: rasa.mladenovic@med.pr.ac.rs; Tel.: +381-69-530-2256
†   These authors contributed equally to this work.

**Abstract:** The aim of our study was to determine the effectiveness of a mobile 3D augmented reality (AR) simulator for local anesthesia training as a serious game. We present a mobile simulator which has three modes for learning: study, 3D simulation, and AR reality. Both pre- and post-training surveys contained open-ended and Likert-scale questions (comprising five response options) on demographics and students' experiences. The response rate was 90.1%. Of the total participants, 37 subjects were female and 27 were male. The mean age was 20 years. The results of the pre-training survey showed that over 80% of dental students from both universities agreed that they were excited about using the mobile simulator. The results from the post training survey showed that 78.7–88.2% of participants felt comfortable using the mobile application, over 72% agreed that it was user friendly, and over 82.3% of participants agreed that it looked realistic. It was also found that 76.6–88.2% of participants agreed that the 3D anatomical structures improved their understanding of LA administration. A serious game for learning local anesthesia can be a very interesting and valuable learning tool for dental students.

**Keywords:** dental education; dental simulator; dental students; local anesthesia; 3D simulation; augmented reality

## 1. Introduction

Dentistry is a profession that demands both intellectual and manual dexterity skills from the operator. Therefore, training dental students is a complex process, as it requires attainment of knowledge and preclinical and clinical training [1]. Generally, dental students receive theoretical knowledge, followed by simulation-based training. A series of structured simulation-based activities based on the hierarchy of needs of the dental students are embedded into the curriculum. Traditionally, phantom heads are used as dental simulators to provide simulation-based training in a conventional preclinical laboratory. The concept of the phantom head or mannequin was first introduced into dentistry by Oswald Fergus in 1894. The administration of profound local anesthesia (LA) is a key skill for dentists and a prerequisite for most dental treatments. Traditionally, dental students have been trained in the skill of administering LA by injecting oranges, peer-to-peer administration, video demonstrations of these procedures, and LA manikin practice before advancing to patients. However, each of these models has its limitations.

As technology continues to advance over time, new types of learning modalities have emerged, including educational tools based on augmented reality (AR), virtual reality

(VR), or a combination of both [2]. In addition, the literature shows that experiences in AR and simulations improve preparedness for dental students [3,4]. Serious games describe practice for education and training, rather than merely games for entertainment purposes. Serious games could be used as an interactive, entertaining, and motivational approach to learning, and they have the potential to enhance the motor and decision-making skills of the user [5]. Due to these advantages, serious games are gaining popularity and are increasingly commonly used in higher education. Recently, there has been a plethora of literature on embedding serious games into healthcare education; however, the evidence on their impacts is limited.

Digital LA simulation devices have been sparsely used in dental education, and they have not been utilized at all for teaching LA administration [1]. In addition, there is limited research about using such devices to improve the student's overall learning experience and skills for administrating LA to a patient. This indicates that further investigation is warranted, and addressing this gap in knowledge forms the focus of this study. We aimed to determine the effectiveness of a mobile 3D and AR simulator for LA training. The serious game is meant to improve the dental student's LA administration technique, experience and confidence. The study addressed two primary research questions: (a) whether the use of the digital LA simulation as a serious game could be beneficial for students, and (b) how learning from the digital LA simulations compared with learning with a traditional LA manikin.

## 2. Materials and Methods

### 2.1. Application of a Serious Game for LA

This study presents the use of the mobile application Dental Simulator (Campinas, SP 13083765, Brazil), which is available for Android (Google Play Store) (Figure 1A–D). The Dental Simulator application has three modes: (1) study mode, (2) simulation, and (3) augmented reality. The study modes include (a) technique descriptions—reading material, where the student can read, learn and recall the theory of the technique; (b) a clinical video showing simple LA procedures; (c) a simulation video of the technique; and (d) practice mode, where the student can practice the technique with transparent skin. Furthermore, the simulation mode allows the student to practice the procedure without transparency, making this mode close to an actual procedure. Finally, the augmented reality mode enables students to download the trackers, and the device's camera can plot the 3D models using VR glasses.

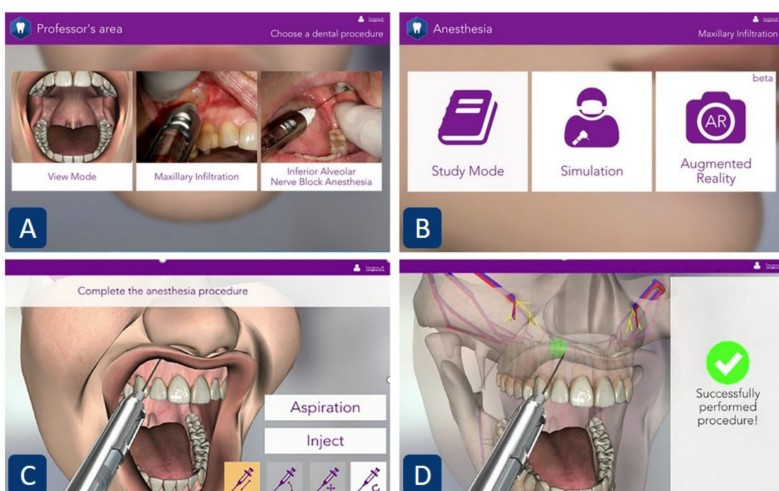

**Figure 1.** The Dental Simulator mobile application. (**A**) Demonstration of various LA administration techniques offered by the application. (**B**) Various modes of the application. (**C**) Demonstrating the syringe control buttons that allow the syringe to rotate, move, reset and inject. (**D**) The demonstration of correct technique for the maxillary infiltration procedure.

### 2.2. Ethical Approval and Participant Recruitment

Ethical approval was obtained from the Human Research Ethics Committee at The University of Queensland (UQ) and the Institutional Review Commission of the Faculty of Medicine, the University of Pristina (UP) (approval number 2019002946). The study involved the completion of a pre-training survey and post-training survey by second-year dental students enrolled in a Bachelor of Dental Science (Honours) at UQ and fourth-year dental students in a five-year study program at UP. The entire cohort of dental students from these years was recruited for this study. The participant consent and information sheet, along with the surveys, were distributed to students who agreed to participate in the study. Within the dental curriculum, dental students in the second year receive a total of 18 h of dental LA training. This includes 6 h of didactic lectures, a 3 h tutorial, 3 h of preclinical training and 6 h of clinical training. During the preclinical training session, students completed LA training on traditional phantom heads (Local Anaesthesia Manikin Model, OneDental, Miami, Australia) for 30 min. They completed the pre-training survey before the Dental Simulator training. This was followed by 30 min of Dental Simulator training and the completion of a post-training survey. The program enables students to perform an inferior alveolar nerve block and supraperiosteal buccal infiltration (Figure 1A). The software has various modes for each procedure (Figure 1B). It also allows students to open the mouth using their fingers, and they can also rotate the model and zoom in or zoom out to place the target point at the correct position. The syringe control buttons allow the syringe to rotate, move, reset, and inject (Figure 1C). It also informs students whether the techniques are correct (turns green) or incorrect (turns red), and injection target sites are also visible for practice. Students first received the demonstration for both techniques, and then they worked in pairs on their smartphones (Figure 1D).

### 2.3. Survey Instrument and Piloting

The pre- and post-training surveys contained open-ended and Likert-scale questions comprising five response options (strongly agree, agree, neutral, disagree, and strongly disagree). The surveys gathered demographics and students' experiences of the Dental Simulator training. We previously validated the survey [1,6]. Surveys were separated from the consent forms and collected anonymously into a designated survey box to ensure confidentiality.

### 2.4. Statistical Analysis

The data were tabulated on a Microsoft Excel spreadsheet and then imported into IBM SPSS Statistics for Macintosh v26 (IBM, Armonk, NY, USA) for descriptive analysis and GraphPad PRISM 9.0 software (GraphPad Software, San Diego, CA, USA) for collation and creation of appropriate graphs. Responses were summarized, and comparisons were made. The outputs of data are presented in a table format as percentages and in graphical format. Specific data tests performed included descriptive statistics, such as frequencies and percentages.

## 3. Results

### 3.1. Participants' Demographics

A total of 71 students were invited to participate in the study; however, only 64 (UQ = 47; UP = 17) students agreed to participate in the pre- and post-training surveys. Hence, the response rate of the study was 90.1%. Of the total participants, 37 subjects were female and 27 were male. The ages of the participants ranged from 19 to 24 years. The mean age was 20 years. Regarding regular use of devices, 93.6% of participants used a smartphone, 6.3% used a smartwatch, 36.1% used a tablet or iPad, and 91.4% used a laptop. Regarding previous experience with augmented or virtual reality, only 6.3% of students had experienced it. The students' expectations before using the Dental Simulator were assessed through a series of statements with which students indicated their agreement or lack thereof (Table 1).

**Table 1.** Pre-training perceptions of dental students.

| Statement | Responses (%) | | |
|---|---|---|---|
| | Agreement | Neutral | Disagreement |
| *I am excited about using the LA dental simulator for LA training* | | | |
| • UQ | 80.85 | 19.15 | 0 |
| • UP | 82.36 | 11.76 | 5.88 |
| *I expect the LA dental simulator to be user friendly* | | | |
| • UQ | 76.6 | 17.02 | 6.38 |
| • UP | 82.35 | 17.65 | 0 |
| *I expect working on the LA dental simulator is realistic* | | | |
| • UQ | 64.96 | 23.4 | 10.64 |
| • UP | 74.47 | 17.65 | 5.88 |
| *I expect that the LA dental simulator will assist my learning* | | | |
| • UQ | 82.98 | 14.89 | 2.13 |
| • UP | 76.47 | 17.65 | 5.88 |
| *I expect that using the LA dental simulator will improve my LA administration skills* | | | |
| • UQ | 85.1 | 10.64 | 4.26 |
| • UP | 94.24 | 11.76 | 0 |
| *I expect working on the LA dental simulator will allow me to see the 3D anatomical structures that will increase my understanding for LA administration* | | | |
| • UQ | 87.24 | 12.77 | 0 |
| • UP | 82.36 | 17.65 | 0 |
| *I expect added value in the use of the LA dental simulator in my training compared to relying solely on traditional methods of training of LA training* | | | |
| • UQ | 82.96 | 17.02 | 0 |
| • UP | 82.36 | 17.65 | 0 |

The University of Queensland (UQ); University of Pristina (UP).

### 3.2. Pre-Training Survey

The responses were grouped as definitely negative (strongly disagree, disagree), neutral, and definitely positive (agree, strongly agree) to better differentiate the clearly negative sentiments. The results of the pre-training survey showed that over 80% of dental students from both universities agreed that they were excited about using the Dental Simulator mobile application and expected Dental Simulator to be user-friendly (Table 1 and Figure 2).

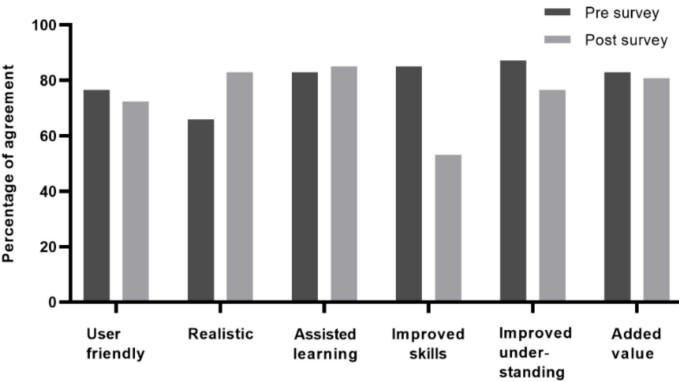

**Figure 2.** Comparison of pre- and post-training survey responses.

Over three quarters of participants perceived that it would assist their learning: 82.9% of participants from UQ and 76.4% of participants from UP. Almost 65% of the participants expected that Dental Simulator would be realistic. Eighty-two percent of participants from both universities agreed that it would add value compared with traditional learning methods. Above 87% of participants agreed that it would improve their LA administration skills. Similarly, over 82% of participants were in agreement with the statement that seeing the 3D anatomical structures through Dental Simulator increased their understanding of LA administration.

*3.3. Post-Training Survey*

After using the Dental Simulator, the students' perceptions were assessed through the post-training survey (Table 2 and Figure 3). The results show that 78.7–88.2% of participants agreed that they felt comfortable using the mobile application, and over 72% agreed it was user friendly. Furthermore, over 82.3% of participants agreed that it looked realistic. It was also found that 76.6–88.2% of participants agreed that the 3D anatomical structures improved their understanding of LA administration, and more than half of the participants (UQ: 53%; UP: 64.7%) felt that it improved their skills. The results also demonstrate that 64.7–80.8% of the participants agreed that it added value in training compared with relying solely on traditional methods, and over 85.1% agreed that it assisted their learning (Figure 3).

**Table 2.** Post-training perceptions of dental students.

| Statement | Responses (%) | | |
|---|---|---|---|
| | **Agreement** | **Neutral** | **Disagreement** |
| *I felt comfortable using the LA dental simulator* | | | |
| • UQ | 78.72 | 19.15 | 2.13 |
| • UP | 88.23 | 11.76 | 0 |
| *Local anesthesia dental simulator is user friendly* | | | |
| • UQ | 72.34 | 19.15 | 8.51 |
| • UP | 76.47 | 23.53 | 0 |
| *3-D images of the anatomy in the LA dental simulator looked realistic* | | | |
| • UQ | 82.98 | 17.02 | 0 |
| • UP | 82.36 | 17.65 | 0 |
| *Using LA dental simulator assisted my learning* | | | |
| • UQ | 85.1 | 12.77 | 2.13 |
| • UP | 88.23 | 11.76 | 0 |
| *I felt more confident about my LA administration skills after using LA dental simulator* | | | |
| • UQ | 59.57 | 34.04 | 6.38 |
| • UP | 64.71 | 17.65 | 17.64 |
| *3-D anatomical structures on LA dental simulator improved my understanding of anatomical landmarks* | | | |
| • UQ | 76.6 | 21.28 | 2.13 |
| • UP | 88.24 | 11.76 | 0 |
| *The use of LA dental simulator added value in my training compared to relying solely on traditional methods* | | | |
| • UQ | 80.85 | 17.02 | 2.13 |
| • UP | 64.71 | 23.53 | 11.76 |

**Table 2.** *Cont.*

| Statement | Responses (%) | | |
|---|---|---|---|
| | **Agreement** | **Neutral** | **Disagreement** |
| The use of LA dental simulator improved my skills of LA administration | | | |
| ● UQ | 53.19 | 31.91 | 14.89 |
| ● UP | 64.71 | 17.65 | 17.65 |
| When using LA dental simulator, I felt I was engaged in a learning activity | | | |
| ● UQ | 78.72 | 19.15 | 2.13 |
| ● UP | 87.58 | 11.76 | 0 |
| I think the use of LA dental simulator would be helpful in teaching LA administration technique | | | |
| ● UQ | 82.98 | 14.89 | 2.13 |
| ● UP | 88.24 | 5.88 | 5.88 |

The University of Queensland (UQ); University of Pristina (UP).

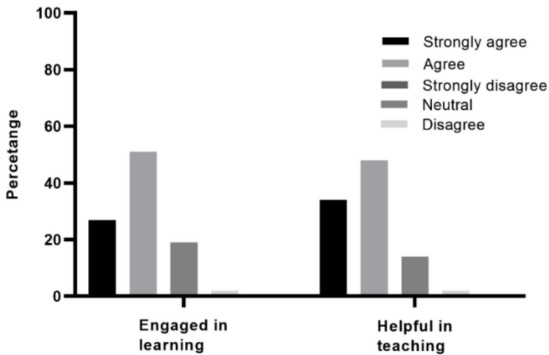

**Figure 3.** Responses to the statements: "When using the LA dental simulator, I felt engaged in a learning activity", and "I think the use of LA dental simulator would be helpful in teaching LA administration technique".

*3.4. Open-Ended Responses*

Students' comments regarding learning included:

"This is very suitable for our learning in the current global pandemic situation".

"Love the idea of the software that we can use in our own time too".

"The fact that the app provides unlimited attempts without any complicated equipment, and we can use it on our device makes it attractive".

"This app has multiples features to explore and it is fun using it". Comments regarding technical difficulties included "Maxillary infiltration was not very accurate in the simulation, the alveolar bone didn't seem to be considered properly", and "the app crashed on my phone, but probably it is my phone".

## 4. Discussion

The findings of the study showed the positive impact of the mobile 3D and AR simulator for LA training. It was perceived as an enjoyable, flexible teaching tool for improving the dental student's LA administration technique, experience, and confidence. In addition, the study indicated that serious games for learning LA could provide interactive ways of presenting material, and therefore should be used as adjunct learning tools in addition to traditional teaching. This additional learning tool could help students develop and practice skills and abilities—e.g., problem solving, critical thinking, and clinical skills.

Teaching methods are changing worldwide and are moving away from traditional lectures toward more integrated courses, where theoretical knowledge and clinical skills are taught simultaneously. Dental education has evolved over the years with advancements in technology and communication. Various dental simulations are being developed and

incorporated into the dental curriculum to support the acquisition of manual dexterity skills before real-life clinical scenarios. Furthermore, the COVID-19 pandemic has presented an array of unique challenges to students and educators. For many months in the previous two years, dental students could not access simulation laboratory facilities, and thus face–face teaching became challenging worldwide. Simulator-based serious games provide students with economical and flexible alternative learning environments.

Dental schools currently use realistic patient simulators with built-in dental models to simulate dental treatment. They allow educators to demonstrate techniques aimed at improving students' hand-eye coordination and manual dexterity [7–9]. Moreover, during the activities on the simulators, dentistry students need constant feedback from teachers about their work to better understand the procedures before moving on to the next procedures [10].

Our 3D pediatric model for dental LA VR uses case-based scenarios related to various aspects of LA administration, and the model showed improved student engagement and learning [1]. Furthermore, a series of studies by Mladenovic et al. [11–13] have explored the use of the mobile simulator (Dental Simulator v1.13 for iOS and Android) on a mobile phone during the pandemic (from home). The use of such pedagogical tools enhance and support learning and increase flexibility for students. They also increase active learning and engagement in an online and blended environment [1]. However, there are no cross-institutional studies to determine the effect of this simulation model. This study addressed that issue by performing trials at two different dental schools. We showed that serious games can improve knowledge among dental students (analyzing results from two universities), as there was an increase in pre-and post-assessment scores. This outcome is similar to those of serious games for other areas of health education [14,15].

Administration of LA is the prerequisite for many dental procedures, and clinicians attempt to master the skills for administrating relatively painless injections, as this is a key to having a cooperative patient [16]. A recent study by Mladenovic et al. [13] concluded that the use of AR might improve students' control of the syringe during their first LA injection to pediatric patients but did not reduce their anxiety levels. Another study investigated the use of the mixed-reality adult model for teaching dental hygienists and demonstrated increased participant confidence [17]. The results of this study are in line with several other studies that showed students supported the use of a 3D learning simulation system for LA techniques over traditional teaching methods [6,13,18,19]. Furthermore, there was no significant difference between the participants from the two dental schools (UQ and UP) regarding agreement or disagreement in the pre- and post-training surveys.

Serious games are designed to simulate authentic learning situations, enabling students to gain experience in safe learning environments without harming patients. Moreover, every mistake in a game can increase the awareness of the student before clinical practice. The fun features of video games can also be built in to enhance engagement and motivation to learn, including a nice interface and use of music and high-quality graphics. Improving engagement and motivation can be considered a key strength of serious games compared to traditional approaches to learning [20]. The significance of these models lies in their ability to provide automatic feedback to students on their performances and in their allowing of multiple attempts. These models are based on the premise that students will acquire their dispositional skills through the linear process of "explore–play–create–together". Our model addresses a gap in existing digital dental technology by providing a 3D interactive model that can enhance students' learning and skills. Furthermore, it offered a safe alternative approach with which students could get hands-on-experience on LA during the COVID-19 pandemic. It does not require patient contact and can be used both on-campus and off-campus. Thus, the model allows access from the comfort of one's own home, it is user-friendly, it provides unlimited attempts to perform tasks, and it provides feedback on performance.

The study showed no significant differences between the participants who regularly used smartphones, smartwatches, tablets/iPad, and laptops compared to non-regular

users. However, one of the study's limitations was that it did not assess whether the skills learned in the virtual learning environment were transferrable to the clinical environment. Furthermore, the virtual learning environment provided by the application does not cater for all types of student, being especially unsuitable for those who take time to become proficient in the use of modern technology. In the future, the application could be improved by employing enhanced utilization of haptic feedback (this allows the device to vibrate when the needle comes in contact with the bone). Furthermore, more realistic anatomical details, involuntary movements of the patient, and the option of choosing the patient's age could also be incorporated. This feature is critical in managing inferior alveolar nerve block in children, as the position of the mandibular foramen differs significantly from that of adults [19].

## 5. Conclusions

A serious game teaching LA to dental students can be interesting and valuable learning tool. It can be used in addition to traditional LA teaching methods, as it increased student engagement and enjoyment; however, limitations with the software persist as barriers. It does not cater for all types of student learning.

**Author Contributions:** R.M. and S.Z. conceived of the ideas, developed the study design, and wrote the manuscript; S.A. and C.P. supervised; K.M. and C.P. proofread the manuscript; all authors critically revised the manuscript. All authors have read and agreed to the published version of the manuscript.

**Funding:** This research received no external funding.

**Institutional Review Board Statement:** Implementation of this study was approved by the Human Research Ethics Committee at The University of Queensland, and by the Institutional Review Commission of the Faculty of Medicine, the University of Pristina in Serbia.

**Informed Consent Statement:** Informed consent was obtained from all subjects involved in the study.

**Data Availability Statement:** Data sharing does not apply to this article, as no datasets were generated during the current study.

**Conflicts of Interest:** The authors declare no conflict of interest.

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
