# Peer review of "Assessing the Pedological Impact of Local Anesthesia Dental Simulator as Serious Game"

_applsci, doi:10.3390/app12073285_

Round 1
Reviewer 1 Report
This research is under the scope of this journal; the topic is relevant to readers, and this research is about potentially significant insights in the field.
The topic is current and it is very interesting and enjoyable to read. Missing comparing this technique with traditional teaching methods, monitoring over time etc...
Still I find it interesting enough to post once they add some of these limitations to the discussion.
I recommend that the English be reviewed by a native
Author Response
REVIEWER 1
This research is under the scope of this journal; the topic is relevant to readers, and this research is about potentially significant insights in the field.
Comment 1: The topic is current and it is very interesting and enjoyable to read. Missing comparing this technique with traditional teaching methods, monitoring over time etc...Still I find it interesting enough to post once they add some of these limitations to the discussion.
- Response: Thank you for your comments. We have added the limitations to the discussion as suggested.
Comment 2: I recommend that the English be reviewed by a native
- Response: The manuscript is now reviewed by the native English speakers and grammar errors have been fixed.
Reviewer 2 Report
When I decided to review the article according to the submitted abstract, I was convinced that the authors had developed their system and conducted research on its basis.
After reading the entire article, I was disappointed with its content. The article’s structure is good, but the information is scientifically very modest. The authors analyzed the application available for mobile devices. They decided to survey how users perceive this application. In my opinion, this is not what research should be like.
The article does not have a good literature review, and the discussion also leaves much to be desired.
After conducting the survey, the authors concluded that the mobile application might be an interesting educational tool in addition to traditional methods.
Such a conclusion can be drawn without conducting any research at all. It is simply obvious.
The article is submitted for a Special Issue: Advances in Augmented Reality, Virtual Reality, and Computer Graphics. Unfortunately, I am not satisfied with the quality of this article, and I believe that it should not be published in this SI in this form.
Author Response
REVIEWER 2
Comment 1: When I decided to review the article according to the submitted abstract, I was convinced that the authors had developed their system and conducted research on its basis. After reading the entire article, I was disappointed with its content. The article’s structure is good, but the information is scientifically very modest. The authors analyzed the application available for mobile devices. They decided to survey how users perceive this application. In my opinion, this is not what research should be like. The article does not have a good literature review, and the discussion also leaves much to be desired.
- Response: We appreciate your comments. However, the aim of the paper was to determine the effectiveness of the mobile 3D and AR simulator for LA training as a serious game in improving the dental student’s LA administration technique, experience and confidence. The main aim of this application was to be simple and easy to cater the needs of all students. As our previous work with digital dentistry showed that not all students are tech-savvy, and some takes time to become proficient in the use of modern technology. This model allows students accessibility from the comfort of their own home, user-friendly, unlimited attempts to perform tasks, and provide feedback on their performance.
Comment 2: After conducting the survey, the authors concluded that the mobile application might be an interesting educational tool in addition to traditional methods. Such a conclusion can be drawn without conducting any research at all. It is simply obvious.
- Response: We have modified our conclusions as suggested
Comment 3: The article is submitted for a Special Issue: Advances in Augmented Reality, Virtual Reality, and Computer Graphics. Unfortunately, I am not satisfied with the quality of this article, and I believe that it should not be published in this SI in this form.
- Response: We have edited our paper as per your suggestions.
Reviewer 3 Report
In this study, Zafar et al. studied the use of the mobile 3D simulation and augmented reality simulator for local anesthesia training as a serious game. It was found that the majority of the participants 78.7-88.2% of participants agreed that they felt comfortable using the mobile application, over 72% agreed it was user-friendly, and over 82.3% of participants were in agreement that it looked realistic. In addition, 76.6-88.2% of participants were in agreement that the 3D anatomical structures improved their understanding of LA administration. This is an interesting study, and it shows the use of modern technology in teaching and learning. Some revisions are needed.
The abbreviations should be written in full form when used for the first time. Such as AI, LA, etc.
“serious games” Why the serious word is used? Or can it be removed or use an alternative word?
“The digital LA simulation devices have been sparsely used in dental education, and particularly it has not been utilized as an adjunct in teaching LA administration.” Better to add Ref?
Materials and methods
“The entire cohort of dental students per year was recruited, thus the sample size calculation was not performed.” Better to add a total number of students and remove the part “thus the sample size calculation was not performed?.
Results
Figures caption should be below the Figure.
In Tables 1 and 2, UQ and UP should be abbreviated below the Table.
Discussion
More discussion is needed. Discussion of the present results is needed. Need to compare the results from this study similar or related studies.
Was there a difference in the results obtained between previous experience and not having experience?
Author Response
REVIEWER 3
In this study, Zafar et al. studied the use of the mobile 3D simulation and augmented reality simulator for local anesthesia training as a serious game. It was found that the majority of the participants 78.7-88.2% of participants agreed that they felt comfortable using the mobile application, over 72% agreed it was user-friendly, and over 82.3% of participants were in agreement that it looked realistic. In addition, 76.6-88.2% of participants were in agreement that the 3D anatomical structures improved their understanding of LA administration. This is an interesting study, and it shows the use of modern technology in teaching and learning. Some revisions are needed.
Comment 1: The abbreviations should be written in full form when used for the first time. Such as AI, LA, etc.
- Response: We have updated all the abbreviations as suggested.
Comment 2: “serious games” Why the serious word is used? Or can it be removed or use an alternative word?
- Response: The term “serious games” is widely accepted terminology in education, and in an educational setting “serious games” is a purposeful learning environment that targets key curriculum areas for explicit learning. Serious games are games or game-like interactive systems developed with game technology and design principles for a primary purpose other than pure entertainment. In this study, the Serious Games-based Learning comparison gives authors the opportunity to research how these applications can be purposefully integrated into the dental curriculum to support student learning. This also enable authors to reflect on the ways in which these serious games can be used to engage students and develop a range of skills and teaching approaches that best support these tools for learning. These are the main reason why we used the terminology “serious games” in this paper.
Comment 3: “The digital LA simulation devices have been sparsely used in dental education, and particularly it has not been utilized as an adjunct in teaching LA administration.” Better to add Ref?
- Response: Reference added for the above statement.
Comment 4: Materials and methods: “The entire cohort of dental students per year was recruited, thus the sample size calculation was not performed.” Better to add a total number of students and remove the part “thus the sample size calculation was not performed?
Response: We have updated the methods as suggested.
Comment 5: Results: Figures caption should be below the Figure.
- Response: We have moved the captions below the figures.
Comment 6: In Tables 1 and 2, UQ and UP should be abbreviated below the Table.
- Response: We have added the abbreviation below the table.
Comment 7: Discussion: More discussion is needed. Discussion of the present results is needed. Need to compare the results from this study similar or related studies.
- Response: We have added to the discussion section as suggested by the reviewer.
Comment 8: Was there a difference in the results obtained between previous experience and not having experience?
- Response: We have added a comment related to the above statement.
Round 2
Reviewer 2 Report
Unfortunately, my opinion of the article has not changed. Nevertheless, it is difficult for the authors to correct it in a few days. Looking at the fact that the other reviewers considered the article good and the Editor approved the paper for improvement, I can only say that the authors responded to other reviewers' comments correctly. Therefore, my opinion should not be taken into account.